# The association between body mass index and all-cause mortality in Japanese patients with incident hemodialysis

**Koji Inagaki**[ORCID]*, **Naoto Tawada**[◉], **Masahiro Takanashi**[◉], **Toshiyuki Akahori**[◉]

Department of Nephrology, Chutoen General Medical Center, Shizuoka, Japan

◉ These authors contributed equally to this work.
* koji-i@chutoen-hp.shizuoka.jp

**Data Availability Statement:** All relevant data are within the manuscript and its Supporting information files.

## Abstract

### Background

A higher body mass index (BMI) has been associated with better survival among chronic kidney disease patients in some reports. However, more research is required to determine the associations between BMI and mortality in incident hemodialysis patients. Therefore, this study aimed to investigate the association between mortality and BMI measured at the first hemodialysis session and 2 weeks after hemodialysis initiation in Japanese patients with incident hemodialysis.

### Methods

We retrospectively enrolled 266 adult patients with incident hemodialysis who were treated at our hospital between May 2013 and June 2019. The data on BMI was obtained at the first hemodialysis session and 2 weeks after hemodialysis initiation. Patients were divided into tertiles based on BMI [<18.5 (low), 18.5–23.9 (normal), and $\geq$24 (high) kg/m$^2$]. The normal group was used as the reference group. The primary outcome was all-cause mortality.

### Results

The mean age of patient was 68.9 ± 12.0 years, and the BMI was 23.3 ± 4.24 kg/m$^2$ at the first hemodialysis session. The body mass index was 22.0 ± 3.80 kg/m$^2$ at 2 weeks after hemodialysis initiation. During a mean follow-up of 3.89 ± 2.12 years, 80 (30.1%) deaths occurred. In multivariate analyses, low BMI at the first hemodialysis session was significantly associated with worse all-cause mortality (hazard ratio, 2.39; 95% confidence interval, 1.13–5.03). At 2 weeks after hemodialysis initiation, high BMI was significantly associated with better all-cause mortality (hazard ratio, 0.38; 95% confidence interval, 0.18–0.81).

### Conclusion

At 2 weeks after HD initiation, high BMI was associated with lower mortality in Japanese patients with incident hemodialysis.

**Funding:** The authors received no specific funding for this work.

**Competing interests:** The authors have declared that no competing interests exist.

## Introduction

Obesity is a major public health problem and is increasing worldwide [1]. Obesity is associated with a higher risk of diabetes, cardiovascular disease (CVD), and cancer, and an increased mortality in the general population [2, 3]. Obesity also increases the risk of chronic kidney disease (CKD) and its progression [4].

However, obesity is inversely associated with better survival among patients with chronic heart failure, chronic obstructive pulmonary disease and older age (i.e., elderly) [5–7]. This phenomenon is referred to as the "obesity paradox" [8]. Obesity appears to be associated with better survival among patients with pre-dialysis CKD and hemodialysis (HD) [9–11].

Among Japanese patients undergoing HD, low body mass index (BMI) at the start of HD was independently associated with an increased risk of all-cause death [12]. However, a higher BMI was not associated with a decreased risk of all-cause death. In addition, Johansen, et al. [13] reported that a high BMI at the beginning of dialysis was not associated with increased survival in Asian patients. Hoogeveen et al. [14] reported that BMI at the start of dialysis was not associated with mortality in elderly dialysis patients. We believe that this finding could be related to volume overload. Thus, we hypothesized if the effects of fluid overload were eliminated, high BMI patients would have a better prognosis. In the present study, we investigated the association between mortality and BMI measured at the first HD session and 2 weeks after HD initiation in Japanese patients with incident HD.

## Materials and methods

### Study design and participants

We conducted a retrospective cohort study in a single center. The participants were 284 patients, aged ≥20 years, who initiated maintenance HD from May 2013 to June 2019. Eighteen patients were excluded because the follow-up period was <3 months. Therefore, 266 patients were analyzed. This study was approved by the Ethics committee of Chutoen General Medical Center (Shizuoka, Japan). The approval number is KENI 139. The study was conducted in accordance with the ethical principles of the Declaration of Helsinki. The Ethics Committee approved a waiver of informed consent because this study was a retrospective study. Instead, we gave eligible patients the opportunity to opt out by posting a disclosure document on our institution's website (https://www.chutoen-hp.shizuoka.jp/department/clinical-research/).

### Clinical parameters

A diagnosis of "diabetes mellitus" ("DM") was defined as a HbA1c levels ≥6.5% or treatment with antidiabetic medications. "Cardiovascular disease" ("CVD") was defined as myocardial infarction, angina requiring percutaneous coronary intervention, hospitalization for heart failure, stroke, or peripheral artery disease requiring revascularization or amputation surgery. "Late referral" was defined as a referral to nephrologist <3 months before dialysis initiation. A temporary catheter was examined at the initiation of HD. An arteriovenous fistula (AVF) was examined at the time of decannulation of the temporal catheter. "Hypoxemia" was defined as a blood oxygen saturation of <90% or treatment with oxygen inhalation. Data on hypoxemia and pedal edema were obtained at the first HD session and 2 weeks after HD initiation. Blood pressure was measured before starting the dialysis. The cardiothoracic ratio was measured on x-ray images before starting the dialysis. The data on blood pressure and cardiothoracic ratio were obtained at the time of first HD session and 2 weeks after HD initiation. Blood tests were conducted on samples obtained at the first HD session and 2 weeks after HD initiation. We

calculated the estimated glomerular filtration rate (eGFR) by using the modified equation for Japanese:

$$eGFR = 194 * sCr^{-1.094} * (age)^{-0.287} * 0.739 \ (\text{if female})$$

[15].

The data of eGFR were obtained only at the first HD session to know the kidney function before HD initiation. Medication use referred to the drugs being taken at the first HD session.

## BMI

The BMI was calculated based on body weight in kilograms divided by height in meters squared. Body weight was defined as 'dry weight' measured after HD session. The data on body weight was obtained at the first HD session and 2 weeks after HD initiation. This is because the state of overhydration was controlled by 2 weeks after HD initiation in most patients [16]. Patients were categorized into three groups: "high" was defined as a BMI $\geq$24 kg/m$^2$; "normal," as a BMI of 18.5–23.9 kg/m$^2$; and "low," as a BMI <18.5 kg/m$^2$ [17].

## Outcome

The primary outcome of our study was all-cause mortality. Patient survival was censored at kidney transplantation, loss to follow-up, or at the end of follow-up in February 2022.

## Statistical analyses

Normally distributed variables are expressed as the mean ± the standard deviation (SD) and were compared by using analysis of variance (ANOVA). Nonparametric variables are expressed as the median and interquartile range (IQR) and were compared by using the Kruskal–Wallis test. Categorical variables are expressed as a number and proportion and were compared by using Fisher's exact test. Kaplan–Meier analysis with log-rank tests was used for the survival analysis of three BMI categorical groups.

We constructed Cox proportional hazards models to determine the risk of all-cause mortality associated with the BMI categories and clinical characteristics. Of the three BMI categorical groups, the normal group was used as the reference group. Model 1 was adjusted for age. Model 2 was adjusted for the variables in Model 1 plus sex and diabetes. Model 3 was adjusted for the variables in Model 2 with the addition of variables with p values < 0.05 in the univariate analysis. We also examined these relationships in the analysis of an elderly subgroup ($\geq$65 years). The results are expressed as hazard ratios (HRs) with 95% confidence intervals (CIs).

All statistical analyses were conducted using EZR (Saitama Medical Center, Jichi Medical University, Saitama, Japan). EZR is a graphical user interface for R statistics (The R Foundation for Statistical Computing, Vienna, Austria) [18].

# Results

## Baseline characteristics

The patients' characteristics are shown in Tables 1 and 2 at the first HD session and 2 weeks after HD initiation, respectively. The mean age was 68.9 ± 12.0 years, and 183 (68.8%) patients were elderly. Overall, 177 (66.5%) patients were male, 144 (54.1%) patients had a history of DM, and 83 (31.2%) patients had a history of CVD. The mean BMI at the first HD session was 23.3 ± 4.24 kg/m$^2$. Overall, 96 (36.1%) patients were in the high BMI group, 148 (55.6%) patients were in the normal BMI group, and 22 (8.3%) patients were in the low BMI group. At 2 weeks after HD initiation, the mean BMI was 22.0 ± 3.80 kg/m$^2$, 71 (26.7%) patients had

**Table 1. Baseline clinical characteristics in 266 patients at the first HD session.**

| variables | All (n = 266) | BMI (The first HD session) ≥24 (n = 96) | 18.5–23.9 (n = 148) | <18.5 (n = 22) | P value |
|---|---|---|---|---|---|
| Age, years | 68.9 ± 12.0 | 62.7 ± 12.9 | 71.6 ± 10.0 | 76.9 ± 8.07 | <0.01 |
| Elder (≥65 years) | 183 (68.8) | 50 (52.1) | 113 (76.4) | 20 (90.9) | <0.01 |
| Male/ Female | 177 (66.5)/ 89 (33.5) | 69 (71.9)/ 27 (28.1) | 97 (65.5)/ 51 (34.5) | 11 (50.0)/ 11 (50.0) | 0.14 |
| Smoking history | 161 (60.5) | 63 (65.6) | 87 (58.8) | 11 (50.0) | 0.33 |
| DM | 144 (54.1) | 64 (66.7) | 74 (50.0) | 6 (27.3) | <0.01 |
| CVD | 83 (31.2) | 25 (26.0) | 52 (35.1) | 6 (27.3) | 0.31 |
| Late referral | 43 (16.2) | 15 (15.6) | 23 (15.5) | 5 (22.7) | 0.65 |
| BMI (first HD session), kg/m$^2$ | 23.3 ± 4.24 | 27.8 ± 3.28 | 21.3 ± 1.57 | 17.3 ± 0.89 | <0.01 |
| BMI (two weeks after HD initiation), kg/m$^2$ | 22.0 ± 3.80 | 25.9 ± 3.10 | 20.2 ± 1.60 | 16.9 ± 0.91 | <0.01 |
| temporary catheter | 76 (28.6) | 27 (28.1) | 41 (27.7) | 8 (36.4) | 0.70 |
| AVF | 235 (88.4) | 87 (90.6) | 132 (89.2) | 16 (72.7) | 0.071 |
| Hypoxemia | 45 (16.9) | 16 (16.7) | 26 (17.6) | 3 (13.6) | 0.97 |
| Pedal edema | 155 (58.3) | 66 (68.8) | 78 (52.7) | 11 (50.0) | 0.030 |
| sBP, mmHg | 155.3 ± 27.6 | 163.4 ± 26.8 | 151.0 ± 26.4 | 149.5 ± 32.0 | <0.01 |
| dBP, mmHg | 79.4 ± 15.5 | 84.7 ± 16.2 | 76.5 ± 14.6 | 75.0 ± 13.3 | <0.01 |
| Cardiothoracic rate, % | 54.6 ± 6.98 | 55.3 ± 7.15 | 54.2 ± 6.91 | 54.5 ± 6.76 | 0.48 |
| Hemoglobin, g/dL | 9.40 ± 1.51 | 9.37 ± 1.49 | 9.44 ± 1.53 | 9.25 ± 1.49 | 0.84 |
| Albumin, g/dL | 3.07 ± 0.63 | 3.03 ± 0.69 | 3.07 ± 0.61 | 3.24 ± 0.42 | 0.38 |
| Potassium, mEq/L | 4.46 ± 0.85 | 4.42 ± 0.80 | 4.50 ± 0.88 | 4.35 ± 0.85 | 0.62 |
| Adjusted Ca, mg/dL | 8.77 ± 0.92 | 8.69 ± 0.88 | 8.79 ± 0.96 | 9.05 ± 0.76 | 0.25 |
| P, mg/dL | 6.29 ± 1.69 | 6.49 ± 1.59 | 6.15 ± 1.74 | 6.40 ± 1.68 | 0.28 |
| Cr, mg/dL | 9.11 [7.73–10.5] | 9.54 [8.41–11.1] | 8.82 [7.46–10.2] | 8.03[6.14–10.8] | <0.01 |
| eGFR, mL/min /1.73 m$^2$ | 4.89 ± 1.55 | 4.76 ± 1.51 | 4.88 ± 1.41 | 5.54 ± 2.40 | 0.11 |
| Use of RAASI | 143 (53.8) | 48 (50.0) | 84 (56.8) | 11 (50.0) | 0.55 |

Values are presented as mean (± SD), median [IQR], and Numbers (%). Abbreviations; HD, hemodialysis; BMI, body mass index; DM, diabetes mellitus; CVD, cardiovascular disease; AVF, Arteriovenous fistula; sBP, systolic blood pressure; dBP, diastolic blood pressure; Ca, calcium; P, phosphorus; Cr, creatine; eGFR, estimated glomerular filtration rate; RAASI, renin-angiotensin-aldosterone system inhibitor.

high BMI, 146 (54.9%) patients had normal BMI, and 49 (18.4%) patients had low BMI groups. The number of patients with hypoxemia decreased from 45 (16.9%) patients at the first HD session to 3 (1.1%) patients at 2 weeks after HD initiation. The number of patients with pedal edema decreased from 155 (58.3%) patients at the first HD session to 34 (12.8%) patients at 2 weeks after HD initiation and that of patients with non-overhydration (no hypoxemia and no pedal edema) increased from 100 patients (37.6%) at the first HD session to 230 patients (86.5%) at 2 weeks after HD initiation. The mean sBP was 155.3 ± 27.6 mmHg at the first HD session and decreased to 146.4 ± 25.5 mmHg at 2 weeks after HD initiation. The mean cardiothoracic rate was 54.6 ± 6.98% at the first HD session and decreased to 52.4 ± 6.57% at 2 weeks after HD initiation.

The characteristics of the three BMI categorical groups at the first HD session are shown in Table 1. Significant differences between the three groups were observed in age, history of DM, pedal edema, systolic blood pressure (sBP), diastolic blood pressure (dBP), and serum creatinine (Cr).

**Table 2. Baseline clinical characteristics in 266 patients at 2 weeks after HD initiation.**

| Variables | All (n = 266) | BMI (Two weeks after HD initiation) | | | P value |
|---|---|---|---|---|---|
| | | ≥24 (n = 71) | 18.5–23.9 (n = 146) | <18.5 (n = 49) | |
| Age, years | 68.9 ± 12.0 | 61.4 ± 12.7 | 70.1 ± 10.6 | 75.9 ± 9.05 | <0.01 |
| Elder (≥65 years) | 183 (68.8) | 35 (49.3) | 105 (71.9) | 43 (87.8) | <0.01 |
| Male/ Female | 177 (66.5)/ 89 (33.5) | 48 (67.6)/ 23 (32.4) | 107 (73.3)/ 39 (26.7) | 22 (44.9)/ 27 (55.1) | <0.01 |
| Smoking history | 161 (60.5) | 46 (64.8) | 93 (63.7) | 22 (44.9) | 0.049 |
| DM | 144 (54.1) | 49 (69.0) | 76 (52.1) | 19 (38.8) | <0.01 |
| CVD | 83 (31.2) | 15 (21.1) | 51 (34.9) | 17 (34.7) | 0.10 |
| Late referral | 43 (16.2) | 10 (14.1) | 25 (17.1) | 8 (16.3) | 0.88 |
| BMI (first HD session), kg/m$^2$ | 23.3 ± 4.24 | 28.6 ± 3.43 | 22.4 ± 2.04 | 18.5 ± 1.30 | <0.01 |
| BMI (two weeks after HD initiation), kg/m$^2$ | 22.0 ± 3.80 | 27.1 ± 2.76 | 21.0 ± 1.39 | 17.4 ± 0.83 | <0.01 |
| temporary catheter | 76 (28.6) | 20 (28.2) | 42 (28.8) | 14 (28.6) | 1.0 |
| AVF | 235 (88.4) | 65 (91.5) | 130 (89.0) | 40 (81.6) | 0.23 |
| Hypoxemia | 3 (1.1) | 0 (0.0) | 0 (0.0) | 3 (6.1) | <0.01 |
| Pedal edema | 34 (12.8) | 13 (18.3) | 18 (12.3) | 3 (6.1) | 0.15 |
| sBP, mmHg | 146.4 ± 25.5 | 151.5 ± 23.4 | 144.7 ± 26.2 | 144.1 ± 25.5 | 0.14 |
| dBP, mmHg | 76.2 ± 13.4 | 80.8 ± 14.4 | 75.6 ± 13.1 | 71.6 ± 11.0 | <0.01 |
| Cardiothoracic rate, % | 52.4 ± 6.57 | 53.1 ± 7.32 | 51.7 ± 6.34 | 53.4 ± 5.97 | 0.19 |
| Hemoglobin, g/dL | 9.58 ± 1.39 | 9.81 ± 1.41 | 9.59 ± 1.43 | 9.19 ± 1.15 | 0.051 |
| Albumin, g/dL | 3.03 ± 0.59 | 3.16 ± 0.67 | 2.99 ± 0.57 | 2.94 ± 0.47 | 0.067 |
| Potassium, mEq/L | 3.99 ± 0.54 | 4.14 ± 0.59 | 3.97 ± 0.50 | 3.86 ± 0.54 | 0.012 |
| Adjusted Ca, mg/dL | 9.07 ± 0.60 | 9.09 ± 0.58 | 9.02 ± 0.64 | 9.18 ± 0.48 | 0.24 |
| P, mg/dL | 4.66 ± 1.27 | 4.91 ± 1.68 | 4.62 ± 1.13 | 4.41 ± 0.87 | 0.086 |
| Cr, mg/dL | 7.07 [5.96–8.58] | 8.02 [6.78–10.0] | 7.03 [6.02–8.40] | 5.97 [5.32–6.98] | <0.01 |
| eGFR, mL/min /1.73 m$^2$ | 4.89 ± 1.55 | 4.72 ± 1.37 | 4.95 ± 1.52 | 4.97 ± 1.89 | 0.57 |
| Use of RAASI | 143 (53.8) | 35 (49.3) | 78 (53.4) | 30 (61.2) | 0.44 |

Values are presented as mean (± SD), median [IQR], and Numbers (%). Abbreviations; HD, hemodialysis; BMI, body mass index; DM, diabetes mellitus; CVD, cardiovascular disease; AVF, Arteriovenous fistula; sBP, systolic blood pressure; dBP, diastolic blood pressure; Ca, calcium; P, phosphorus; Cr, creatine; eGFR, estimated glomerular filtration rate; RAASI, renin-angiotensin-aldosterone system inhibitor.

The characteristics of the three BMI categorical groups at 2 weeks after HD initiation are shown in Table 2. Significant differences between the three groups were observed in age, sex, smoking history, history of DM, hypoxemia, dBP, serum potassium, and serum Cr.

## Mortality

During a mean follow-up of 3.89 ± 2.12 years, 80 (30.1%) deaths occurred. Of the 80 deaths, 22 (27.5%) were due to cardiac disease, 20 (25.0%) were due to infectious disease, 12 (15.0%) were due to malignancy, 7 (8.8%) were due to stroke. The 1-year mortality rate was 5.8%, and 3-year mortality rate was 16.2%. Among 183 elderly dialysis patients, 70 (38.3%) deaths occurred. The 1-year mortality rate was 7.3% and the 3-year mortality rate was 21.1% in elderly dialysis patients. Fig 1 shows the Kaplan–Meier curves for the cumulative survival rates of the three BMI categorical groups in the first HD session and 2 weeks after HD initiation. At the first HD session, patients in low BMI group had significantly worse survival than did patients in the other two groups. The 3-year mortality rate was 10.5% in the high BMI, 16.4% in the normal group, and 40.4% in the low BMI group. However, at 2 weeks after HD initiation, patients in

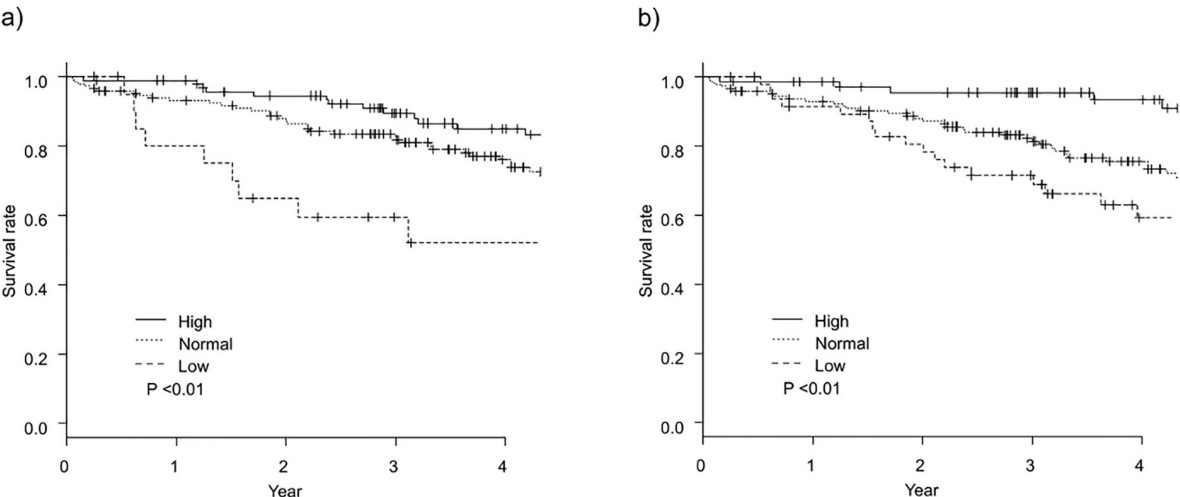

**Fig 1. Kaplan–Meier survival curve analyses of event-free survival, based on the BMI category.** a) The first HD session. b) Two weeks after HD initiation. Abbreviations; BMI; body mass index, HD; hemodialysis.

the high BMI group had significantly better survival than did patients in the other two groups. The 3-year mortality rate was 4.5% in the high BMI group, 17.7% in the normal group, and 28.5% in the low BMI groups.

## HRs on univariate and multivariate analysis

In univariate Cox regression analysis, age, history of CVD, AVF, hypoxemia, sBP, eGFR, and BMI category were significantly associated with survival outcome at the first HD session. At 2 weeks after HD initiation, age, history of CVD, AVF, hypoxemia, sBP, albumin, cardiothoracic rate, eGFR and BMI category were found to be significantly associated with survival outcome in univariate Cox regression analysis. At the first HD session, high (HR, 0.49; 95% CI, 0.29–0.85) and low BMI (HR, 2.78; 95% CI, 1.51–5.15) were significantly associated with all-cause mortality. At 2 weeks after HD initiation, high (HR, 0.31; 95% CI, 0.15–0.63) and low BMI (HR, 1.74; 95% CI, 1.06–2.87) were significantly associated with all-cause mortality (Tables 3 and 4).

We adjusted these statistics in Model 1 (i.e., age), Model 2 (i.e., Model 1 plus sex and DM), and Model 3 (i.e., Model 2 plus CVD, AVF, hypoxemia, sBP, and eGFR) (Table 3). In Models 1, 2, and 3, low BMI (Model 3; HR, 2.39; 95% CI, 1.13–5.03) was significantly associated with worse all-cause mortality at the first HD session. High BMI was not significantly associated with survival outcome in Models 1, 2, and 3 at the first HD session.

Similarly, we adjusted these statistics in Model 1 (i.e., age), Model 2 (i.e., Model 1 plus sex and DM), and Model 3 (i.e., Model 2 plus CVD, AVF, hypoxemia, sBP, albumin, cardiotho-racic rate, and eGFR) at 2 weeks after HD initiation (Table 4). At 2 weeks after HD initiation, high BMI (Model 3: HR, 0.38; 95% CI: 0.18–0.81) was significantly associated with better all-cause mortality in Models 1, 2, and 3. Low BMI was not significantly associated with survival outcome in Model 1, 2, and 3 at 2 weeks after HD initiation.

In elderly (≥65 years old) patients, low BMI (Model 3: HR, 2.35; 95% CI:1.08–5.12) was also significantly associated with worse all-cause mortality at the first HD session (Table 3). At 2 weeks after HD initiation, high BMI (Model 3: HR, 0.23; 95% CI:0.09–0.61) was significantly

**Table 3. Hazard ratio of BMI category by multivariate Cox regression analysis for all-cause mortality in all HD patients and elderly (≥65 years) HD patients at the first HD session.**

| Variables | All<br>HR (95% CI) | Elderly<br>HR (95% CI) |
|---|---|---|
| Model 0 | | |
| High | 0.49 [0.29–0.85] | 0.54 [0.29–0.997] |
| Normal | 1.0 (reference) | 1.0 (reference) |
| Low | 2.78 [1.51–5.15] | 2.50 [1.34–4.67] |
| Model 1 | | |
| High | 0.69 [0.40–1.20] | 0.61 [0.32–1.14] |
| Normal | 1.0 (reference) | 1.0 (reference) |
| Low | 1.93 [1.03–3.62] | 1.97 [1.04–3.76] |
| Model 2 | | |
| High | 0.68 [0.39–1.18] | 0.60 [0.32–1.13] |
| Normal | 1.0 (reference) | 1.0 (reference) |
| Low | 2.35 [1.21–4.57] | 2.34 [1.18–4.64] |
| Model 3 | | |
| High | 0.72 [0.40–1.29] | 0.64 [0.33–1.24] |
| Normal | 1.0 (reference) | 1.0 (reference) |
| Low | 2.39 [1.13–5.03] | 2.35 [1.08–5.12] |

Model 0: unadjusted, Model 1: adjusted age, Model 2: Adjusted for Model 1 plus sex and DM, Model 3: Adjusted for Model 2 plus CVD, AVF, hypoxemia, sBP, and eGFR. Abbreviations; BMI, body mass index; HD, hemodialysis; HR, hazard ratio; CI, confidence interval; DM, diabetes mellitus; CVD, cardiovascular disease; AVF, Arteriovenous fistula; sBP, systolic blood pressure; eGFR, estimated glomerular filtration rate.

**Table 4. Hazard ratio of BMI category by multivariate Cox regression analysis for all-cause mortality in all HD patients and elderly (≥65 years) HD patients at 2 weeks after HD initiation.**

| Variables | All<br>HR (95% CI) | Elderly<br>HR (95% CI) |
|---|---|---|
| Model 0 | | |
| High | 0.31 [0.15–0.63] | 0.23 [0.09–0.59] |
| Normal | 1.0 (reference) | 1.0 (reference) |
| Low | 1.74 [1.06–2.87] | 1.62 [0.96–2.71] |
| Model 1 | | |
| High | 0.39 [0.19–0.81] | 0.26 [0.10–0.65] |
| Normal | 1.0 (reference) | 1.0 (reference) |
| Low | 1.30 [0.78–2.17] | 1.31 [0.76–2.26] |
| Model 2 | | |
| High | 0.40 [0.19–0.82] | 0.26 [0.10–0.68] |
| Normal | 1.0 (reference) | 1.0 (reference) |
| Low | 1.44 [0.84–2.45] | 1.43 [0.82–2.51] |
| Model 3 | | |
| High | 0.38 [0.18–0.81] | 0.23 [0.09–0.61] |
| Normal | 1.0 (reference) | 1.0 (reference) |
| Low | 1,43 [0.81–2.53] | 1.46 [0.79–2.70] |

Model 0: unadjusted, Model 1: adjusted age, Model 2: Adjusted for Model 1 plus sex and DM, Model 3: Adjusted for Model 2 plus CVD, AVF, hypoxemia, sBP, albumin, cardiothoracic rate, and eGFR. Abbreviations; BMI, body mass index; HD, hemodialysis; HR, hazard ratio; CI, confidence interval; DM, diabetes mellitus; CVD, cardiovascular disease; AVF, Arteriovenous fistula; sBP, systolic blood pressure; eGFR, estimated glomerular filtration rate.

associated with better all-cause mortality. The HRs of all-cause death in high BMI were lower in elderly patients at 2 weeks after HD initiation in Models 1, 2, and 3 (Table 4).

## Discussion

In this study, we evaluated the association between BMI and all-cause mortality in Japanese patients with incident HD. We found that low BMI was associated with higher all-cause mortality at the first HD session. High BMI was not significantly associated with all-cause mortality at the first HD session; however, it was associated with lower all-cause mortality at 2 weeks after HD initiation. These associations were stronger in elderly HD patients.

The higher mortality risk for low BMI patients at the start of dialysis is in keeping with the findings of other studies. De Mutsert et al. [19] reported that a low BMI (i.e., <18.5 kg/m$^2$) was associated with higher mortality risk in Dutch dialysis patients. Toida et al. [12] reported that low BMI (i.e., <18.5 kg/m$^2$) increased the risk of all-cause mortality in Japanese dialysis patients. Two reasons may explain why a lower BMI increases the risk of all-cause mortality. First, low BMI can result from low food intake due to poor appetite and dietary restrictions. HD patients with malnutrition have a lower BMI, which is associated with higher mortality [20]. In a Japanese nationwide cohort study, malnutrition has been reported as a risk factor for death in HD patients [21]. In our study, BMI decreased 2 weeks after HD initiation because the volume overload was corrected. Thus, the low BMI group at the first HD session was actually leaner. Second, patients with a low BMI are more likely to have a low muscle mass. Carrero et al. [22] reported that muscle atrophy was more prominent in patients with a lower BMI and was associated with higher mortality in HD patients. Zhang et al. [23] reported that frailty significantly increased the mortality risk in end-stage renal disease patients. Patel et al. [24] reported that serum creatine was a surrogate marker of muscle mass in HD patients. In this study, low BMI patients had a lower serum Cr at 2 weeks after HD session. Thus, their low muscle mass may have been associated with the low BMI.

In contrast, low BMI at 2 weeks after HD initiation was not significantly associated with survival outcome in the multivariate analyses. We assumed that there was a large difference in lean body mass between low and normal BMI groups at HD initiation. However, there was a small difference in lean body mass between low and normal BMI groups at 2 weeks after HD initiation. Lean body mass is critical for skeletal muscle mass assessment [25]. Additionally, a low lean body mass is an independent risk factor for malnutrition [26]. Arase et al. [27] reported that a low lean body mass was associated with an increased risk for all-cause mortality in Japanese patients on maintenance HD. Out of 49 patients in the low BMI group at 2 weeks after HD initiation, 28 (57.1%) belonged to the normal BMI group at the first HD session. We speculated that these patients would not have been excessively restricted nutritionally before HD initiation. As a result, even if these patients had undergone dialysis for 2 weeks to correct their fluid overload, they would have lost little muscle mass. Therefore, future studies need to investigate lean body mass in incident HD patients.

In our study, the association of higher BMI with lower death risk was more pronounced in elderly HD patients. Similarly, Nagai et al. [28] reported that a high BMI was associated with an improved all-cause mortality in elderly patients undergoing HD, but not in middle-aged patients undergoing HD, in Japan. We speculated that muscle and fat mass have a greater effect on mortality in elderly patients than in young patients. Liu et al. reported that lean body mass had a greater effect on all-cause mortality in older adults than in younger adults [29]. Additionally, there was a monotonic positive association of fat mass with all-cause mortality in younger participants, but an approximate J-shaped trend in older participants [29]. In this study, high BMI groups in elderly patients had a higher serum Cr at 2 weeks after HD session

than the other two groups. Additionally, only one of the 35 elderly patients in high BMI groups had a BMI >30 kg/m2. We speculate that older patients with high BMI had adequate muscle and fat mass. Therefore, adequate nutrition and exercise are important in elderly patients with pre-dialysis CKD.

This study had several limitations. First, because the study population was small, we could not completely adjust for known prognostic factors. In addition, some clinical factors result in wide CIs. Second, because this study was observational study, we could not rule out residual confounding factors. Third, BMI does not distinguish between fat and muscle mass. Thus, higher BMI may imply a high fat mass or a high muscle mass. BMI had been measured only at the time of HD initiation in incident HD patients in other studies [13, 30–32]. however, in the current study, the BMI was measured at the first HD session and 2 weeks after HD initiation. Measuring BMI at 2 weeks after HD initiation could eliminate the influence of volume over-load. In addition, measuring the BMI twice could make clinicians aware of the change in BMI from the HD initiation period.

## Conclusions

In conclusion, low BMI at the first HD session was associated with higher all-cause mortality. High BMI at 2 weeks after HD initiation was associated with lower all-cause mortality. This tendency was stronger in elderly HD patients. In Japanese patients with incident HD, adequate nutrition and physical activity might have improved the prognosis.

## Supporting information

**S1 Table. Anonymous data set of 266 patients.**
(XLSX)

## Acknowledgments

We are grateful to Ryoko Sakamoto, Takako Toyoda, Wakako Sato, Taeko Kurita, and Shinzo Kitajima for their contributions to this study.

## Author Contributions

**Conceptualization:** Koji Inagaki.

**Data curation:** Koji Inagaki, Naoto Tawada, Masahiro Takanashi.

**Formal analysis:** Koji Inagaki.

**Investigation:** Naoto Tawada, Masahiro Takanashi, Toshiyuki Akahori.

**Methodology:** Koji Inagaki.

**Project administration:** Koji Inagaki.

**Supervision:** Koji Inagaki, Toshiyuki Akahori.

**Writing – original draft:** Koji Inagaki, Naoto Tawada, Masahiro Takanashi, Toshiyuki Akahori.

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
