## [Decision Letter · Decision Letter 0]

21 Feb 2022

PONE-D-21-33831The association between body mass index and all-cause mortality in Japanese patients with incident hemodialysis

PLOS ONE

Dear Dr. Inagaki,

Thank you for submitting your manuscript to PLOS ONE. After careful consideration, we feel that it has merit but does not fully meet PLOS ONE’s publication criteria as it currently stands. Therefore, we invite you to submit a revised version of the manuscript that addresses the points raised during the review process.

The reviewers commented favorably on your manuscript, but had some worthwhile suggestions. The authors should address the remaining issues.

We look forward to receiving your revised manuscript.

Kind regards,

Michinari Nakamura, MD

Academic Editor

PLOS ONE

Journal Requirements:

2.  Please provide additional details regarding participant consent. In the ethics statement in the Methods and online submission information, please ensure that you have specified what type you obtained (for instance, written or verbal, and if verbal, how it was documented and witnessed). If your study included minors, state whether you obtained consent from parents or guardians. If the need for consent was waived by the ethics committee, please include this informatio

Reviewers' comments:

Reviewer's Responses to Questions

**Comments to the Author**

1. Is the manuscript technically sound, and do the data support the conclusions?

Reviewer #1: Partly

2. Has the statistical analysis been performed appropriately and rigorously? 

Reviewer #1: No

3. Have the authors made all data underlying the findings in their manuscript fully available?

Reviewer #1: Yes

4. Is the manuscript presented in an intelligible fashion and written in standard English?

Reviewer #1: Yes

5. Review Comments to the Author

Reviewer #1: Please find the attached the comments file.

1)The conclusions could not be drawn appropriately based on the data presented.

2)The method of data collection and statistical analysis are unclear.

There are two baselines.

6. PLOS authors have the option to publish the peer review history of their article (what does this mean?). If published, this will include your full peer review and any attached files.

Reviewer #1: **Yes: **Sawako Kato

---

## [Author Response · Author response to Decision Letter 0]

1 Apr 2022

Point-by-point responses to Reviewer’s comment

We would like to thank the reviewers for the time and effort in reviewing our manuscript and providing comments and suggestions, which have considerably helped us improve our manuscript. We have answered each of your points below and hope that our responses and revisions address all your comments.

Comment 1: Which BMI did use to divide patients into three groups, at the first time HD session or 2 weeks after? Or, did the authors conduct survival analysis on two types of patient groups one from the other divided into three BMI categories? Did the authors do one survival analysis based on the data sheet divided into three groups by the first time BMI and another survival analysis based on the data sheet divided into three by the BMI after 2 weeks? If so, the authors should indicate not only the baseline data after 2 weeks of dialysis (Table1), but also the data after the start of dialysis. And The two survival analyzes should have different observation periods of 14 days. In particular, They should confirm whether the baseline data changes or not, since it was mentioned that the number of people in the three groups differs depending on when BMI was measured.

Response 1: All data except for BMI were obtained at the fist HD session, and we had included the data at the first HD session in Table 1 in the original version of the manuscript. We have added the baseline data from the first hemodialysis (HD) session in Table 1 in revised version of the manuscript. We searched clinical and laboratory data from 2 weeks after HD initiation and have added baseline data at 2 weeks after HD initiation in Table 2. We have also provided detailed definitions and measurements of the factors in the “Clinical parameters” section (from page 5 line 78 to page 6 line 99).

 We reconstructed Cox proportional hazards models to determine the risk of all-cause mortality associated with the BMI categories and clinical characteristics based on data in the first HD session and 2 weeks after HD initiation. As the result, univariate Cox regression analysis revealed that age, history of CVD, AVF, hypoxemia, sBP, albumin, cardiothoracic rate, eGFR and BMI category were significantly associated with survival outcome at 2 weeks after HD initiation.

Because there were too many variables with p values <0.05 in the univariate analysis, we extended the observational period from August 2021 to February 2022. Five patients who were excluded due to the follow-up period being <3 months in the original version of the manuscript were added in the revised version because we were able to monitor them for >3 months. Based on this comment, data were added for six patients who died within 3 months. As a result, the number of patients increased from 255 in the original version of the paper to 266 in the revised version. Similarly, the number of deaths in patients increased from 68 in the original version of the paper to 80 in the revised version.

In multivariate analyses, low BMI at the first hemodialysis session was significantly associated with worse all-cause mortality (hazard ratio, 2.39; 95% confidence interval, 1.13–5.03). At 2 weeks after the hemodialysis session, high BMI was significantly associated with better all-cause mortality (hazard ratio, 0.38; 95% confidence interval, 0.18–0.81) (from page 2 line 35 to page 3 line 39). In elderly (≥65 years) patients, low BMI (Model 3: HR, 2.35; 95% CI: 1.08–5.12) was also significantly associated with worse all-cause mortality at the first HD session. At 2 weeks after HD initiation, high BMI (Model 3: HR, 0.23; 95% CI: 0.09–0.61) was significantly associated with better all-cause mortality. Thus, the revised manuscript has similar results as the previous manuscript. We have described these results in HRs on univariate and multivariate analysis section (From page 16 line 200 to page 21 line 251).

Comment 2: The authors stated that “the objective of the present study was to investigate the association between BMI and mortality in Japanese incident hemodialysis patients before and after the treatment of volume overload”. (line 66-68) and that “in our study, BMI decreased 2 weeks after HD initiation because the volume overload was corrected”. (line 246-247) . I agree that most patients probably improved their volume status. But, if they want to compare mortality between before and after the treatment of volume overload, they should evaluate and indicate data to support results that volume overload was corrected.

Were all enrolled patients in condition with volume overload? They should indicate volume status at the first dialysis session.

How many patients reached the adequate body weight after 2 weeks ultrafiltration treatment?.

Response 2: To assess volume overload, pedal edema and hypoxemia has been added in the revised manuscript. At the first HD session, there were 45 (16.9%) patients with hypoxemia and 155 (58.3%) patients with pedal edema. At 2 weeks after HD initiation, there were three (1.1%) patients with hypoxemia and 34 (12.8%) patients with pedal edema. The number of patients with non-overhydration (no hypoxemia and no pedal edema) increased from 97 patients (36.5%) at the first HD session to 230 patients (86.5%) at 2 weeks after HD initiation. Therefore, we have added these results to the revised manuscript in the baseline characteristics section (From page 8 line 144 to page 9 line 154).

Comment 3: The author should show the source to decide “BMI after 2 weeks treatment”

Response 3: Based on your suggestion, we have added the following sentence “This is because the state of overhydration was controlled by 2 weeks after HD initiation in most patients [16]. (Page 6 line 103-104)” in the BMI section. The reference 16 has also been cited (from page 28 line 387 to 389).

Comment 4: Also, the cardiothoracic ratio was relatively large in all groups in the data after 2 weeks of dialysis. They should show the data of cardiothoracic ratio decreased at the start of dialysis?

Response 4: In the original version of the manuscript, cardiothoracic ratio was measured at the first HD session, as shown in Table 1. In the revised manuscript, cardiothoracic ratio in the first HD session has been included in Table 1, and the cardiothoracic ratio at 2 weeks after HD initiation has been included in Table 2. The mean cardiothoracic rate was 54.6 ± 6.98% at the first HD session and decreased to 52.4 ± 6.57% at 2 weeks after HD initiation We have added these values in the baseline characteristics section (from page 9 line 152 to page 153).

Comment 5: Also, which body weight did they use to calculate BMI, before HD session or after HD session? We generally use body weight after HD session as dry weight = body weight without excess volume.

Response 5: In this manuscript, we used body weight after HD session to calculate BMI. Thus, we have added a sentence “Body weight was defined as ‘dry weight’ measured after HD session.” (from page 6 line 102).

.

Comment 6: The authors used “BMI” and “overweight” as the same, but should use them separately. BMI is just a calculated number. High BMI include obesity, volume overload = overhydration, and overweight, and healthy with optimal weight, for instance high muscle mass. The unorganized and mixed terms makes readers confused.

Response 6: According to your comment, we have modified the term “overweight” to “high BMI”. Similarly, we have modified the term “underweight” to “low BMI”.

Comment 7: I could not catch what was the aim of this study. The authors stated that “The objective of the present study was to investigate the association between BMI and mortality in Japanese incident hemodialysis patients before and after the treatment of volume overload”. But they discussed only the association between BMI and mortality while they referred the previous reports.

One clinical implication of this study may be that more attention should be paid to HD patients with low BMI because they are very likely in malnutrition status as compatible many previous reports. Weight loss as a consequence of disease including CKD may induce the early mortality associated with a low BMI. Another implication may be that some HD patients with low BMI could have hidden protein energy waste or frail.

Response 7: We believe that the causes of poor prognosis in the low BMI group at the first HD session are malnutrition and frailty. Thus, we have added the following sentences: “In a Japanese nationwide cohort study, malnutrition has been reported as a risk factor for death in HD patients [21].” (from page 21 line 267 to line 268).

We have also added “Zhang et al. [23] reported that frailty significantly increased the mortality risk in end-stage renal disease patients.” (from page 22 line 273 to line 274). Additionally, we changed the sentence “In Japanese patients with incident HD, adequate nutrition and physical activity might have improved the prognosis.” (from page 25 line 328 to line 330) in Conclusions section.

Comment 8: And the authors should discuss why the association between low BMI after 2 weeks treatment and mortality disappeared.

Response 8: We believe that this is because there are many patients who have lost weight by strictly restricting their diet for a short time. We have added the following sentence “On the other hand, low BMI at 2 weeks after HD initiation was not significantly associated with survival outcome in multivariate analyses. This is possibly because many patients who lost weight by strictly restricting their diet for a short time were included in this study. In CKD stage V patients, fluid overload, fatigue, and electrolyte abnormalities tended to occur because of decreasing urine production. Many patients prevent the occurrence of electrolyte abnormalities and overhydration by dietary restriction [25]. Therefore, it is necessary to investigate the changes in the type and quantity of food intake in pre-dialysis CKD patients in future studies.” (From page 22 line 278 to line 285) in the Discussion section.

Comment 9: The authors should discuss the implication that association of higher BMI with lower death risk was more pronounced in elderly HD patients. Indeed, Japanese HD patients shows better survival that HD patients in Western countries. But it was not enough to explain the association of higher BMI with lower death risk.

Response 9: We believe that the reason of association of higher BMI with lower death may be physical activity. Thus, we have added a sentence “In our study, the association of higher BMI with a lower mortality risk was more pronounced in elderly HD patients. Similarly, Oliva et al. [26] reported that high BMI was associated with decreased mortality in HD patients aged >75 years. We believe the reason for this is low physical activity. Inaguma et al. [27] reported that a low Barthel index, a measure of physical activity, was associated with a higher mortality risk in Japanese patients at HD initiation. In their study, the lower the Barthel index, the older the patient and the lower was the BMI.” (From page 23 line 287 to line 292) in the Discussion section.

Comment 10: I guess that the best survival prospects for incident dialysis patients may be those who need not to lose weight after initiation dialysis therapy because euvolemia, and those who are gain body weight because less uremia. Alternatively, I also guess that if patients are suffer from heart failure, the patients with better cardiac function can remove water more. So the larger lose weight may be better prognosis in patients with heart failure. In this study, I would like to see the data of changes in body weight during the 2 weeks and the association between these change and mortality.

Response 10: We calculated the difference in BMI at the first HD session and 2 weeks after the start of dialysis. The median BMI difference was 1 (IQR, 0.4-2.1). Then, we analyzed the differences in BMI by quartiles. The 3-year mortality rate was 10.9% in Q1, 13.6% in Q2, 18.3% in Q3, and 21.2% in Q4. However, it was not significant (p = 0.94) using Kaplan–Meier analysis with log-rank tests. Moreover, in univariate Cox regression analysis, the HRs in Q2, Q3, and Q4 were 1.07 [0.57-2.00], 0.90 [0.48-1.68], and 0.92 [0.48-1.77], respectively (The reference group was Q1).

Comment 11: Six patients who died within three months were included as exclusion criteria. I think the main purpose of this study is to investigate the mortality of hemodialysis patients, so why did you exclude patients who died within 3 months? And was there any patients who died within 2 weeks?

Response 11: All the six patients died between 2 weeks and 3 months after the start of dialysis. Because they had not been followed up for >3 months, the six dead patients were excluded in the original version of the manuscript. As the reviewer suggested, the six dead patients were included in the analysis in the revised manuscript. However, the addition of these six patients to the analysis did not change the results.

Comment 12: The reference below (line 259-261) was not suitable for discussion about higher BMI with lower death risk. Add another proper one.

Polinder-Bos et al. [22] reported that a lower BMI was associated with an increased 1-year mortality, but it was not associated with increased long-term mortality risk in elderly incident dialysis patients in the United States of America (USA).

Response 12: Based on your comment, we have deleted the concerned sentence in the original version of the manuscript. Instead, we have added the following sentence “Similarly, Vashistha et al. [28] reported that high BMI (≥25 kg/m2) was not associated with lower mortality in HD patients aged >65 years.” (from page 23 line 297 to line 298) in the Discussion section.

Comment 13: The authors had better show cause of deaths.

Response 13: We have added the following sentence “Of the 80 deaths, 22 (27.5%) were due to cardiac disease, 20 (25.0%) were due to infectious disease, 12 (15.0%) were due to malignancy, 7 (8.8%) were due to stroke.” (Page 15 Line 181 to 183) in mortality section.

---

## [Decision Letter · Decision Letter 1]

6 May 2022

PONE-D-21-33831R1The association between body mass index and all-cause mortality in Japanese patients with incident hemodialysisPLOS ONE

Dear Dr. Inagaki,

Thank you for submitting your manuscript to PLOS ONE. After careful consideration, we feel that it has merit but does not fully meet PLOS ONE’s publication criteria as it currently stands. Therefore, we invite you to submit a revised version of the manuscript that addresses the points raised during the review process.

The reviewers commented favorably on your manuscript, but had some worthwhile suggestions. The authors should address the remaining issues, including the discussion on potential mechanism. I am pleased to accept your manuscript, based on your revising it.<o:p></o:p>

We look forward to receiving your revised manuscript.

Kind regards,

Michinari Nakamura, MD

Academic Editor

PLOS ONE

Journal Requirements:

Reviewers' comments:

Reviewer's Responses to Questions

**Comments to the Author**

1. If the authors have adequately addressed your comments raised in a previous round of review and you feel that this manuscript is now acceptable for publication, you may indicate that here to bypass the “Comments to the Author” section, enter your conflict of interest statement in the “Confidential to Editor” section, and submit your "Accept" recommendation.

Reviewer #1: (No Response)

2. Is the manuscript technically sound, and do the data support the conclusions?

Reviewer #1: Yes

3. Has the statistical analysis been performed appropriately and rigorously? 

Reviewer #1: Yes

4. Have the authors made all data underlying the findings in their manuscript fully available?

Reviewer #1: Yes

5. Is the manuscript presented in an intelligible fashion and written in standard English?

Reviewer #1: Yes

6. Review Comments to the Author

Reviewer #1: I would like to second review the manuscript submitted to Plos One by Koji Inagaki et al, Title; “The association between body mass index and all-cause mortality in Japanese patients with incident hemodialysis”.

The manuscript has been much more improved than the previous one.

I thank the authors for corrections.

But there are still some concerns.

Comment & Response 3:

The authors should put the sentence “This is because the state of overhydration was controlled by 2 weeks after HD initiation in most patients [16] after the sentence “The data on body weight was obtained at the first HD session and 2 weeks after HD initiation.”

Comment & Response 6:

We still find the term “overweight” and “underweight” in the manuscript.

Comment & response 8:

The authors did not answer the question “why the association between low BMI after 2 weeks treatment and mortality disappeared”

The patients number of low BMI was increased after 2 weeks after HD initiation. Perhaps, I suppose most patients who lost weight could not eat enough by anorexia by uremia and in part dietary restriction pre and post dialysis initiation.

But, this does not explain the question.

Comment & response 9:

Same as Q8, the authors did not answer the question “why association of higher BMI with lower death risk was more pronounced in elderly HD patients”

7. PLOS authors have the option to publish the peer review history of their article (what does this mean?). If published, this will include your full peer review and any attached files.

Reviewer #1: **Yes: **Sawako Kato

---

## [Author Response · Author response to Decision Letter 1]

27 May 2022

May 26, 2022

Michinari Nakamura, MD, PhD

Academic Editor

Plos One

Dear Dr. Nakamura:

We wish to re-submit the attached manuscript entitled “The association between body mass index and all-cause mortality in Japanese patients with incident hemodialysis”. The manuscript ID is PONE-D-21-33831R1.

 We thank you and the reviewer for your thoughtful suggestions and insights. The manuscript has benefited from these insightful suggestions. I look forward to working with you and the reviewer to move this manuscript closer to publication in the PLoS One.

The manuscript has been rechecked and the necessary changes have been made in accordance with the reviewers’ suggestions. The responses to all comments have been prepared and attached.

Thank you for your consideration. I look forward to hearing from you.

Sincerely,

Koji Inagaki

Department of Nephrology, Chutoen General Medical Center, 

Shizuoka, Japan

E-mail: koji-i@chutoen-hp.shizuoka.jp

Point-by-point responses to Reviewer’s comment

We appreciate the time and efforts of the reviewer in reviewing this manuscript again. We have addressed all issues in the review report and believed that the revised version now meets the journal publication requirements.

Comment & Response 3: The authors should put the sentence “This is because the state of overhydration was controlled by 2 weeks after HD initiation in most patients [16] after the sentence “The data on body weight was obtained at the first HD session and 2 weeks after HD initiation.”

Response: We appreciate the reviewer’s valuable comments. Accordingly, we have changed the sentence as follows:

“The data on body weight was obtained at the first HD session and 2 weeks after HD initiation. This is because the state of overhydration was controlled by 2 weeks of HD initiation in most patients [16].” (page 6, lines 103–105).

Comment & Response 6: We still find the term “overweight” and “underweight” in the manuscript.

Response: We appreciate the reviewer’s careful observation and valuable comments on highlighting this issue. Accordingly, we have changed the terms “overweight” and “underweight” to “high BMI” and “low BMI,” respectively, in subsection “Mortality” of the “Results” section (page 15, lines 180–195).

Comment & response 8: The authors did not answer the question “why the association between low BMI after 2 weeks treatment and mortality disappeared”. The patients number of low BMI was increased after 2 weeks after HD initiation. Perhaps, I suppose most patients who lost weight could not eat enough by anorexia by uremia and in part dietary restriction pre and post dialysis initiation. But, this does not explain the question.

Response: We appreciate the reviewer’s valuable advice. We assumed that there was a large difference in lean body mass between low and normal BMI groups at HD initiation. However, there was a small difference in lean body mass between low and normal BMI groups at 2 weeks after HD initiation. Lean body mass is critical for skeletal muscle mass assessment [25]. Additionally, a low lean body mass is an independent risk factor for malnutrition [26]. Arase et al. [27] reported that a low lean body mass was associated with an increased risk for all-cause mortality in Japanese patients on maintenance HD. Out of 49 patients in the low BMI group at 2 weeks after HD initiation, 28 (57.1 %) belonged to the normal BMI group at the first HD session. We speculated that these patients would not have been excessively restricted nutritionally before HD initiation. As a result, even if these patients had undergone dialysis for 2 weeks to correct their fluid overload, they would have lost little muscle mass. Therefore, future studies need to investigate lean body mass in incident HD patients. We have added these sentences to our manuscript (page 22 line 279 to page 23 line 291) in the Discussion section.

Comment & response 9: Same as Q8, the authors did not answer the question “why association of higher BMI with lower death risk was more pronounced in elderly HD patients”

Response: We appreciate the reviewer’s valuable comment. We assumed that muscle and fat mass have a greater effect on mortality in elderly patients than in young patients. Thus, we revised the sentence as follows:

“In our study, the association of higher BMI with lower death risk was more pronounced in elderly HD patients. Similarly, Nagai et al. [28] reported that a high BMI was associated with an improved all-cause mortality in elderly patients undergoing HD, but not in middle-aged patients undergoing HD, in Japan. We speculated that muscle and fat mass have a greater effect on mortality in elderly patients than in young patients. Liu et al. reported that lean body mass had a greater effect on all-cause mortality in older adults than in younger adults [29]. Additionally, there was a monotonic positive association of fat mass with all-cause mortality in younger participants, but an approximate J-shaped trend in older participants [29]. In this study, high BMI groups in elderly patients had a higher serum Cr at 2 weeks after HD session than the other two groups. Additionally, only one of the 35 elderly patients in high BMI groups had a BMI >30 kg/m2. We speculate that older patients with high BMI had adequate muscle and fat mass. Therefore, adequate nutrition and exercise are important in elderly patients with pre-dialysis CKD” (page 23 line 292 to page 24 line 305) in the Discussion section.

---

## [Editor Report · Decision Letter 2]

30 May 2022

The association between body mass index and all-cause mortality in Japanese patients with incident hemodialysis

PONE-D-21-33831R2

Dear Dr. Inagaki,

We’re pleased to inform you that your manuscript has been judged scientifically suitable for publication and will be formally accepted for publication once it meets all outstanding technical requirements.

Kind regards,

Michinari Nakamura, MD

Academic Editor

PLOS ONE
---

## [Editor Report · Acceptance letter]

16 Jun 2022

PONE-D-21-33831R2 

The association between body mass index and all-cause mortality in Japanese patients with incident hemodialysis 

Dear Dr. Inagaki:

I'm pleased to inform you that your manuscript has been deemed suitable for publication in PLOS ONE. Congratulations! Your manuscript is now with our production department. 

Kind regards, 

on behalf of

Dr. Michinari Nakamura 

Academic Editor

PLOS ONE